# Proposed cut-off points for anthropometric and bioelectrical measures based on overweight and obesity criteria in Spanish institutionalised elderly people

Jose Ramon Alvero-Cruz[1], Rosalia Fernandez Vazquez[1], Javier Martinez Blanco[2], Ignacio Rosety[3], Antonio Jesus Diaz[4], Miguel Angel Rosety[5], Manuel Rosety-Rodriguez[5], Francisco Javier Ordonez[5]*

1 Department of Physiology, Histology, Pathological Anatomy and Sport Sciences, School of Medicine, University of Malaga, Málaga, Spain, 2 Nursing Home Division, Andalusian Health Council, Málaga, Spain, 3 Department of Human Anatomy, School of Medicine, University of Cádiz, Cádiz, Spain, 4 School of Nursing, University of Cádiz, Cádiz, Spain, 5 School of Sports Medicine, University of Cádiz, Cádiz, Spain

* franciscojavier.ordonez@uca.es

## Abstract

The increasing prevalence of obesity among the institutionalised elderly population and its severe consequences on health requires an early and accurate diagnosis that can be easily achieved in any clinical setting. This study aimed to determine new cut-off values for anthropometric and bioelectrical impedance measures that are superior to body mass index criteria for overweight and obesity status in a sample of Spanish institutionalised elderly population. A total of 211 institutionalised older adults (132 women, aged 84.3±7.3 years; 79 men, aged 81.5±7.3 years) were enrolled in the current cross-sectional study. Anthropometric and bioelectrical impedance measures included the body mass index, waist circumference, gluteal circumference, waist-hip ratio, sagittal-abdominal diameter, trunk fat, and visceral-fat ratio. In women, the waist circumference, gluteal circumference, sagittal-abdominal diameter, trunk fat, and visceral-fat index presented strongly significant specificity and sensitivity (area under the curve [AUC], p<0.0001) and elevated discriminative values (receiver operating characteristic [ROC] curves: 0.827 to 0.867) for overweight and obesity status. In men, the waist-hip ratio, waist circumference, gluteal circumference, sagittal-abdominal diameter, trunk fat, and visceral-fat ratio were strongly significant AUC (p<0.0001), with moderate-to-high values (ROC curves: 0.757–0.871). In conclusion, our findings suggest that gluteal circumference, waist circumference, and sagittal-abdominal diameter in women and trunk fat, visceral-fat ratio, and waist circumference in men may represent more suitable cut-off values superior to body mass index criteria for overweight and obesity in the Spanish institutionalised elderly population.

**Data Availability Statement:** All relevant data are within the paper and its Supporting Information files.

**Funding:** The authors received no specific funding for this work.

**Competing interests:** The authors have declared that no competing interests exist.

## Introduction

The prevalence of obesity among older European adults has reached epidemic proportions [1]. In Spain, increasing prevalence of obesity coupled with the growth of this population group has become a matter of major concern in recent decades [2]. Although most studies have focused on community-dwelling elderly [1–3], the prevalence of obesity could be even worse among institutionalised elderly individuals in nursing homes (NH) [4, 5]. In fact, obesity has been reported as a major risk factor for institutionalisation [6].

A recent systematic review has revealed an association between obesity with increased morbidity and functional decline [7]. More specifically, overweight and obesity status have been significantly associated with poor cognitive performance among Spanish institutionalised elderly individuals [8]. Furthermore, the obesity status of institutionalised residents may also have a negative impact on staff, equipment, and services provided in NH [9]. Accordingly, caring for obese residents is more expensive than caring for their normal weight counterparts [10]. In fact, obesity status has been considered a criterion of non-admission in NH when choosing prospective residents [9].

For the reasons mentioned above, the recommendation to maintain stable and healthy weight in later stages of life is widely accepted [11]. In order to achieve this goal, an early diagnosis of overweight and obesity that is easily achievable in the clinical setting may play a key role. In this respect, anthropometric findings could be applied across a large population because of its simplicity, cost-effectiveness, portability, and safety [12]. However, the cut-off values for defining both obesity and overweight are currently based on the adult non-elderly population, although there is an increasing body of evidence suggesting that regional fat distribution changes with aging [12, 13]. More specifically, anthropometric indices that rely on waist circumference (WC) are more accurate and reliable for predicting obesity-related comorbidities in elderly individuals as abdominal fat accumulation increases with aging [14].

Thus, the aim of the present study was to define new cut-off values for anthropometric and bioelectrical impedance measures with respect to body mass index (BMI) based overweight and obesity status in a Spanish institutionalised elderly population. We found that the most suitable diagnostic indicators of obesity in institutionalised elderly individuals were gluteal circumference, WC, and SAD for women and Tfat, VFR, and WC for men.

## Materials and methods

### Participants

A total of 210 individuals participated in a cross-sectional multicentre study of central obesity and metabolic syndrome (132 women and 78 men). The current study was conducted between May 2018 and May 2019. All participants were institutionalised in six public NH in the province of Malaga, in the southeast of Spain. Participants were recruited through contact with NH directors and by directly contacting elderly individuals. All participants received detailed information about the objectives and procedures of the current study. The inclusion criteria were as follows: 1) individuals institutionalised in public NH and 2) ability to sign a free and informed consent form. The exclusion criteria were as follows: 1) patients unable to remain in the supine decubitus position due to comorbidities (heart failure, chronic obstructive pulmonary disease, or balance and mobility disorders); 2) severe physical and/or cognitive impairment; 3) WC greater than 130 cm, a value which prevented reliable determination of abdominal bioelectrical impedance; and 4) difficulty in venous access.

The study was approved by the Ethics Committee Board of the University of Malaga (EME-FYDE report: 017–2019) and the Helsinki principles for human research were respected.

## Anthropometric assessment

Weight was measured on a SECA 813 (Hamburg, Germany) calibrated electronic scale to the nearest 100 g. Height was measured on a SECA 216 wall mounted-stadiometer (Hamburg, Germany) with an accuracy of 0.1 cm.

The anthropometric indices BMI (weight in kg/height in square meters) and waist-hip ratio (WHR) were calculated, whereas the sagittal abdominal diameter (SAD) was obtained in the decubitus position at the umbilical reference with a Holtain anthropometer (Crymych, UK).

The WC was set at the level of the iliac crest. The gluteal circumference at the level of the maximum buttock protrusion and ahead at the pubic symphysis level. All values were measured with an inextensible tape to the nearest 1 mm (Lufkin, model W606PM, Cooper Tools, Mexico). The anthropometric measures were obtained in accordance with the guidelines of the International Society for Advancement in Kinanthropometry (ISAK) and were carried out by the same level 3 ISAK-accredited researcher and with a technical error of measurement of less than 1% for all measures.

## Abdominal bioelectrical impedance

Measurements were obtained using the AB-140 model, ViScan® bioelectrical impedance device (Tanita, Japan). With the subject in supine decubitus position, the WC was initially measured at the umbilical level, according to the instrument protocol and by the manufacturer's guidelines by the projection of light in the coronal plane. The values for trunk fat (Tfat) and visceral fat were obtained after positioning a belt with four electrodes, centrally around the umbilical level. Body composition variables were derived by extrapolating impedance measurements (6.2–50 kHz) defining the Tfat, and are expressed in percentage value (range 0%–75%), whereas the visceral fat level is expressed as the visceral fat ratio (VFR) (in arbitrary units, range 1–59). *It should be pointed out that ViScan device* has previously been reported to be equal to dual energy X-ray absorptiometry (DEXA) in estimating abdominal fat mass [15].

## Definition of overweight and obesity status

Classification of overweight (BMI 25–29.9 kg/m$^2$) and obesity (BMI > 30 kg/m$^2$) status was based on the National Institutes for Health (NIH)/WHO guidelines for BMI classification [16].

## Statistical analysis

A descriptive study was carried out to analyse the data. Continuous variables are presented as mean values ± standard deviation (SD). The Kolmogorov-Smirnov test was used to identify the normal distribution of the data. Receiver operating characteristics (ROC) curves were obtained to select cut-off values by evaluating the areas under the curve (AUC), as optimal measures of predictors of the cut-off vales able to correctly discriminate the high and the low risks of the condition. The optimal cut-off point was selected by maximising Youden's index, which is the difference between the true positive rate (sensitivity) and the false positive rate (1-specificity) in the ROC curve. Finally, the Mann-Whitney test was performed to determine differences between sexes in anthropometric and abdominal bioelectrical impedance variables. All statistical analyses were carried out using the MedCalc program for Windows version 19.4.0 (Ostend, Belgium) and a p-value < 0.05 was indicative of statistically significant differences in all cases.

**Table 1. Comparisons of anthropometric and bioelectrical impedance measures of central obesity between genders in an institutionalised elderly population.**

| Variable | Males | | Females | | $p$ |
|---|---|---|---|---|---|
| | Mean | SD | Mean | SD | |
| **Age (years)** | 81.51 | 7.29 | 84.27 | 7.27 | 0.0087 |
| **Weight (kg)** | 68.57 | 13.96 | 63.37 | 12.98 | 0.0069 |
| **Height (cm)** | 162.52 | 8.39 | 151.84 | 6.94 | < 0.0001 |
| **BMI (kg/m$^2$)** | 26.00 | 5.15 | 27.52 | 5.47 | 0.047 |
| **WHR** | 0.98 | 0.08 | 0.97 | 0.08 | 0.153 |
| **Waist (cm)** | 96.66 | 13.34 | 100.02 | 12.61 | 0.07 |
| **Gluteal (cm)** | 97.88 | 9.26 | 103.45 | 11.27 | 0.0004 |
| **SAD (cm)** | 21.24 | 3.91 | 21.77 | 3.85 | 0.35 |
| **Tfat (%)** | 29.73 | 8.79 | 41.59 | 8.46 | <0.0001 |
| **VFR** | 15.47 | 6.34 | 12.81 | 5.18 | 0.017 |

BMI: Body mass index, WHR: Waist to Hip Ratio, SAD: Sagittal abdominal diameter, Tfat: Trunk fat, VFR: Visceral Fat Ratio. Significance level was set at $p < 0.05$ in all cases.

## Results

Table 1 summarises the sex-based differences in the anthropometric measurements, and highlights significant differences observed in age, weight, height, and BMI (all, $p < 0.05$). There were also differences in WC, gluteal circumference, Tfat, and VFR (all $p < 0.05$). No differences were observed in the WHR and SAD ($p > 0.05$).

### ROC curves defining overweight status

Overweight status is defined as BMI 25–29.9 kg/m$^2$. Among women, the WHR did not show significant differences as a predictor of overweight/obesity weight status in terms of AUC ($p > 0.05$) when compared to BMI. Conversely, the WC, gluteal circumference, SAD, Tfat, and VFR showed significantly different AUC values compared to BMI (all, $p < 0.0001$) with high AUC values (between 0.85 and 0.889) (Table 2). Among men, the WHR, WC, gluteal circumference, waist circumference, gluteal circumference, SAD, Tfat, and VFR presented very

**Table 2. Receiver operating characteristics of anthropometric and bioelectrical impedance of central obesity variables in both genders for overweight (BMI = 25 to 29.9 kg/m$^2$).**

| Gender | ROC curve | WHR | Waist | Gluteal | SAD | Tfat | VFR |
|---|---|---|---|---|---|---|---|
| **Females** | **AUC** | 0.601 | 0.889 | 0.88 | 0.863 | 0.859 | 0.85 |
| | **SE** | 0.0531 | 0.0314 | 0.0341 | 0.0331 | 0.0344 | 0.0387 |
| | **95% CI** | 0.509 to 0.689 | 0.820 to 0.938 | 0.809 to 0.932 | 0.789 to 0.918 | 0.785 to 0.916 | 0.774 to 0.909 |
| | **P** | 0.0562 | <0.0001 | <0.0001 | <0.0001 | <0.0001 | <0.0001 |
| | **Youden index J** | 0.219 | 0.6682 | 0.6445 | 0.5688 | 0.6208 | 0.6197 |
| **Males** | **AUC** | 0.752 | 0.907 | 0.83 | 0.89 | 0.822 | 0.794 |
| | **SE** | 0.0577 | 0.0331 | 0.0473 | 0.0409 | 0.0494 | 0.0525 |
| | **95% CI** | 0.638 to 0.845 | 0.818 to 0.962 | 0.726 to 0.907 | 0.797 to 0.951 | 0.715 to 0.902 | 0.684 to 0.880 |
| | **P** | <0.0001 | <0.0001 | <0.0001 | <0.0001 | <0.0001 | <0.0001 |
| | **Youden index J** | 0.4864 | 0.7329 | 0.5214 | 0.6792 | 0.5608 | 0.5068 |

AUC: Area under curve, SE: Standard error, CI: Confidence Interval, p: p value, WHR: Waist to Hip Ratio, SAD: Sagittal abdominal diameter, Tfat: Trunk fat, VFR: Visceral Fat Ratio. Significance level was set at $p < 0.05$ in all cases.

**Table 3. Sensitivity and specificity of cut-off points for anthropometric and bioelectrical impedance variables in relation to body mass index for overweight criteria in an institutionalised elderly population.**

| Variable | Cut-off | Sensitivity | 95% CI | Specificity | 95% CI | +LR | 95% CI | -LR | 95% CI |
|---|---|---|---|---|---|---|---|---|---|
| WHR | >1 | 40.51 | 29.6 - 52.1 | 81.4 | 66.6 - 91.6 | 2.18 | 1.1 - 4.3 | 0.73 | 0.6 - 0.9 |
| Waist | >96 | 85 | 75.3 - 92.0 | 81.82 | 67.3 - 91.8 | 4.68 | 2.5 - 8.8 | 0.18 | 0.1 - 0.3 |
| Gluteal | >101.7 | 73.75 | 62.7 - 83.0 | 90.7 | 77.9 - 97.4 | 7.93 | 3.1 - 20.3 | 0.29 | 0.2 - 0.4 |
| SAD | >19.2 | 92.59 | 84.6 - 97.2 | 64.29 | 48.0 - 78.4 | 2.59 | 1.7 - 3.9 | 0.12 | 0.05 - 0.3 |
| Tfat | >40.9 | 78.75 | 68.2 - 87.1 | 83.33 | 68.6 - 93.0 | 4.73 | 2.4 - 9.4 | 0.26 | 0.2 - 0.4 |
| VFR | >11 | 81.01 | 70.6 - 89.0 | 80.95 | 65.9 - 91.4 | 4.25 | 2.3 - 8.0 | 0.23 | 0.1 - 0.4 |
| WHR | >0.97553 | 74.29 | 56.7 - 87.5 | 74.36 | 57.9 - 87.0 | 2.9 | 1.6 - 5.1 | 0.35 | 0.2 - 0.6 |
| Waist | >95.5 | 86.11 | 70.5 - 95.3 | 87.18 | 72.6 - 95.7 | 6.72 | 2.9 - 15.4 | 0.16 | 0.07 - 0.4 |
| Gluteal | >97.5 | 77.78 | 60.8 - 89.9 | 74.36 | 57.9 - 87.0 | 3.03 | 1.7 - 5.3 | 0.3 | 0.2 - 0.6 |
| SAD | >21.3 | 81.08 | 64.8 - 92.0 | 86.84 | 71.9 - 95.6 | 6.16 | 2.7 - 14.2 | 0.22 | 0.1 - 0.4 |
| Tfat | >29.8 | 75 | 57.8 - 87.9 | 81.08 | 64.8 - 92.0 | 3.96 | 2.0 - 7.9 | 0.31 | 0.2 - 0.6 |
| VFR | >15 | 75 | 57.8 - 87.9 | 75.68 | 58.8 - 88.2 | 3.08 | 1.7 - 5.6 | 0.33 | 0.2 - 0.6 |

CI: Confidence Interval, LR: Likelihood ratio, WHR: Waist to Hip Ratio, SAD: Sagittal abdominal diameter, Tfat: Trunk fat, VFR: Visceral Fat Ratio. Significance level was set at $p < 0.05$ in all cases.

significant AUC values (all, p< 0.0001), with moderate-to-high values between 0.752 and 0.907 (Table 2).

Table 3 shows the new cut-off values in relation to overweight values defined by BMI and sensitivity (Sens) and specificity (Spec) for both men and women for each predictor evaluated in the study.

In women, the WC, gluteal circumference, SAD, Tfat, and VFR, had moderate-to-high Sens values (74%–93%) and Spec (65%–83%). Similarly, in men, anthropometric variables and indices had moderate-to-high Sens values (74%–86%) and Spec (74%–87%) (Table 3).

## ROC curves associated with obesity (BMI >30 kg/m$^2$)

Obesity status is defined as BMI >30 kg/m$^2$. Among women, the stratification by WHR did not show significant AUC values (p > 0.05) to discriminate obesity status. In contrast, the WC, gluteal circumference, SAD, Tfat, and the VFR presented strongly significant AUC (all, p< 0.0001), with high values between 0.827 and 0.867 (Table 4). Among men, the WHR, WC, gluteal circumference, SAD, Tfat, and the VFR presented strongly significant AUC (all, p< 0.0001), with moderate-to-high predictor ROC curves, between 0.757–0.871 (Table 4).

Table 5 reports the cut-off points in relation to obesity status and the sensitivity (Sens) and specificity (Spec) for both female and male institutionalised elderly individuals for each predictor. Among women, the WC, gluteal circumference, SAD, Tfat, and VFR, showed moderate-to-high Sens values (75–89%) and Spec (66–85%). Among men, anthropometric indices also had moderate-to-high Sens values (86–100%) and Spec (69–93%) (Table 5).

## Discussion

To the best of our knowledge, this is the first study to propose optimal cut-off points for anthropometric and bioelectrical measures corresponding to the BMI criteria for overweight and obesity status in a Spanish institutionalised elderly population. These findings could be of great interest in clinical practice considering that recent studies have emphasised the importance of determining population-specific cut-off values for more accurate techniques, such as computed tomography (CT) or DEXA [17].

**Table 4. Receiver operating characteristics analysis of anthropometric and bioelectrical impedance of central obesity variables in both genders for obesity (BMI ≥ 30).**

| Gender | ROC curve | WHR | Waist | Gluteal | SAD | Tfat | VFR |
|---|---|---|---|---|---|---|---|
| Females | AUC | 0.548 | 0.862 | 0.867 | 0.846 | 0.835 | 0.827 |
| | SE | 0.0621 | 0.039 | 0.0347 | 0.0423 | 0.0392 | 0.0404 |
| | 95% CI | 0.456 to 0.639 | 0.789 to 0.918 | 0.793 to 0.921 | 0.770 to 0.905 | 0.757 to 0.896 | 0.748 to 0.890 |
| | P | 0.4357 | <0.0001 | <0.0001 | <0.0001 | <0.0001 | <0.0001 |
| | Youden index J | 0.1475 | 0.6237 | 0.5903 | 0.6006 | 0.5801 | 0.5477 |
| Males | AUC | 0.757 | 0.868 | 0.853 | 0.861 | 0.871 | 0.869 |
| | SE | 0.0608 | 0.0554 | 0.0655 | 0.0674 | 0.0417 | 0.0406 |
| | 95% CI | 0.644 to 0.849 | 0.770 to 0.935 | 0.752 to 0.924 | 0.762 to 0.930 | 0.772 to 0.938 | 0.770 to 0.937 |
| | P | <0.0001 | <0.0001 | <0.0001 | <0.0001 | <0.0001 | <0.0001 |
| | Youden index J | 0.4667 | 0.6171 | 0.6171 | 0.7916 | 0.6897 | 0.75 |

AUC: Area under curve, SE: Standard error, CI: Confidence Interval, p: p value, WHR: Waist to Hip Ratio, SAD: Sagittal abdominal diameter, Tfat: Trunk fat, VFR: Visceral Fat Ratio. Significance level was set at $p < 0.05$ in all cases.

According to the present study, the most suitable diagnostic indicators of obesity in institutionalised elderly individuals were gluteal circumference, WC, and SAD for women and Tfat, VFR, and WC for men. The WHR showed a lower discriminatory ability to predict obesity compared to the other indicators tested in our study, which was in contrast to previously reports for both women and men. With regard to overweight status, the WC, gluteal circumference, and SAD in women and WC, SAD and gluteal circumference in men showed a greater diagnostic ability than the other parameters tested.

In agreement with previous studies focused on German and Australian adults (aged 20–69 years-old) [18, 19], our findings revealed that there was a stronger correlation between WC and BMI when compared to WHR and BMI. In a recent study, Pinheiro et al. [20] found that the BMI underestimated the fat mass percentage in patients with non-dialysis chronic kidney disease because it was not possible to consider the loss of lean mass concomitant to fat gain.

**Table 5. Sensitivity and specificity of cut-off points for anthropometric and bioelectrical impedance variables in relation to body mass index for obesity criteria in an institutionalised elderly population.**

| Gender | Variable | Cut-off | Sensitivity | 95% CI | Specificity | 95% CI | +LR | 95% CI | -LR | 95% CI |
|---|---|---|---|---|---|---|---|---|---|---|
| Females | WHR | >1.02 | 34.29 | 19.1 - 52.2 | 80.46 | 70.6 - 88.2 | 1.75 | 0.9 - 3.3 | 0.82 | 0.6 - 1.1 |
| | Waist | >103 | 80.56 | 64.0 - 91.8 | 81.82 | 72.2 - 89.2 | 4.43 | 2.8 - 7.1 | 0.24 | 0.1 - 0.5 |
| | Gluteal | >104 | 88.57 | 73.3 - 96.8 | 70.45 | 59.8 - 79.7 | 3 | 2.1 - 4.2 | 0.16 | 0.06 - 0.4 |
| | SAD | >23 | 75 | 57.8 - 87.9 | 85.06 | 75.8 - 91.8 | 5.02 | 2.9 - 8.6 | 0.29 | 0.2 - 0.5 |
| | Tfat | >45 | 77.78 | 60.8 - 89.9 | 80.23 | 70.2 - 88.0 | 3.93 | 2.5 - 6.2 | 0.28 | 0.1 - 0.5 |
| | VFR | >12 | 88.89 | 73.9 - 96.9 | 65.88 | 54.8 - 75.8 | 2.61 | 1.9 - 3.6 | 0.17 | 0.07 - 0.4 |
| Males | WHR | >0.949 | 100 | 76.8 - 100.0 | 46.67 | 33.7 - 60.0 | 1.87 | 1.5 - 2.4 | 0 | |
| | Waist | >96.2 | 92.86 | 66.1 - 99.8 | 68.85 | 55.7 - 80.1 | 2.98 | 2.0 - 4.4 | 0.1 | 0.02 - 0.7 |
| | Gluteal | >98 | 92.86 | 66.1 - 99.8 | 68.85 | 55.7 - 80.1 | 2.98 | 2.0 - 4.4 | 0.1 | 0.02 - 0.7 |
| | SAD | >24.6 | 85.71 | 57.2 - 98.2 | 93.44 | 84.1 - 98.2 | 13.07 | 4.9 - 34.5 | 0.15 | 0.04 - 0.6 |
| | Tfat | >32.9 | 92.31 | 64.0 - 99.8 | 76.67 | 64.0 - 86.6 | 3.96 | 2.4 - 6.4 | 0.1 | 0.02 - 0.7 |
| | VFR | >17 | 100 | 75.3 - 100.0 | 75 | 62.1 - 85.3 | 4 | 2.6 - 6.2 | 0 | |

CI: Confidence Interval, LR: Likelihood ratio, WHR: Waist to Hip Ratio, SAD: Sagittal abdominal diameter, Tfat: Trunk fat, VFR: Visceral Fat Ratio. Significance level was set at $p < 0.05$ in all cases.

This finding could be of particular interest considering the high prevalence of sarcopenia among the institutionalised elderly population in Spain [21]. In fact, in this population group, BMI has been considered a marker of protein stores rather than of adiposity, which may ultimately explain, at least in part, the obesity paradox [22].

It is widely accepted that WC reflects abdominal obesity across the adult lifespan [23, 24]. More specifically, on the basis of our findings we propose that cut-off points for WC of 103 cm (80.5% Sens.; 81.8% Spec.) in women and 96.2 cm (92.8% Sens.; 68.8% Spec.) in men can better discriminate obesity status in the Spanish institutionalised elderly population. WC has been associated with risk factors for cardiovascular disease (CVD) to a greater extent than the BMI or the WHR in the elderly [25]. However, it should be pointed out that WC may overestimate or underestimate the risk of CVD as it does not consider differences in height [18]. In this respect, individuals of shorter height may have higher health risks than taller individuals in the moderately-to-large WC group for both sexes and across different ages among Japanese adults [26]. Furthermore, WC has also been associated with a significantly increased risk of dementia among older adults [27]. Additionally, abdominal obesity (WC > 88 cm) has also been positively associated with fragility fractures in a sample of community-dwelling elderly Israeli women [28]. In fact, central obesity, expressed as increased WC, was the only anthropometric parameter identified with negative effects on both the physical and mental domains of quality of life among community-dwelling older adults [29].

Several studies have pointed out that anthropometric and bioelectrical impedance measures are crucial for cardiometabolic risk assessment [23, 30, 31]. In fact, anthropometric indices have been positively correlated with levels of serum proinflammatory cytokines not only in relatively young older adults (aged 60–80 years) [32], but also in female nursing home residents aged $\geq$ 80 years [30]. Furthermore, several studies have emphasised that the cut-off values for these anthropometric indicators depend on age, sex, and race-ethnicity [23, 30]. In a previous study focusing on the same cohort of institutionalised elderly, we reported the cut-off values able to predict metabolic syndrome using BMI (26.81 and 23.53kg/m$^2$), WC (102 and 91cm), SAD (22.1 and 20.7cm), Tfat (34% and 43.7%), and VFR (17 and 11.5) in men and women, respectively [23].

In particular, in women, the SAD, Tfat, and VFR cut-off values for metabolic syndrome ranged within those for overweight and obesity status. Conversely, the WC and WHR cut-offs for metabolic syndrome were higher than the optimal cut-off values defined for obesity among the institutionalised elderly population. In men, the SAD and VFR cut-offs for metabolic syndrome also ranged within the cut-off values established for overweight and obesity status. Conversely, the WC, gluteal circumference, WHR, and Tfat cut-offs for metabolic syndrome were higher than optimal cut-off values for obesity in the institutionalised elderly population [23].

The present study has numerous strengths. It is the first report to propose optimal cut-off values for anthropometric and bioelectrical indices to discriminate overweight and obesity status among the institutionalised elderly population in Spain. In this respect, the use of standardised procedures for anthropometric and bioelectrical determinations contributed to minimising measurement bias. Notably, the anthropometric data were obtained by level 3 ISAK-accredited technicians provided that the difficulties in collecting WC and height and ensuring their accuracy in elderly population [33]. Lastly, participants were recruited from a well-defined population, which represented a single ethnic group (Caucasian), aged above 65 years, and institutionalised in NH.

Conversely, several limitations to the study should also be recognised. The sample size is small and validation of the findings would benefit from a larger population. In addition, our study population comprised a cross-sectional cohort of institutionalised elderly individuals,

and the results may not apply to the community-dwelling elderly. Finally, the exclusion of patients with WC > 130 cm could also be considered a limitation.

## Conclusions

Our study showed that the gluteal circumference, WC, and SAD in women and Tfat, VFR, and WC in men were better indicators of obesity than other anthropometric and bioelectrical impedance measures in the Spanish institutionalised elderly population. Further multicentre studies with larger sample sizes are required to confirm the predictive value of the current optimal cut-off points of anthropometric and bioelectrical measurements in identifying overweight and obesity in institutionalised elderly. Our findings propose simple indices for accurate diagnosis of overweight and obesity that can be easily performed in a clinical setting, which will contribute to managing the severe consequences on health of this condition among elderly individuals in institutionalised settings.

## Supporting information

**S1 File.**
(XLSX)

## Author Contributions

**Conceptualization:** Jose Ramon Alvero-Cruz, Javier Martinez Blanco, Francisco Javier Ordonez.

**Data curation:** Jose Ramon Alvero-Cruz, Antonio Jesus Diaz, Manuel Rosety-Rodriguez, Francisco Javier Ordonez.

**Formal analysis:** Ignacio Rosety, Antonio Jesus Diaz, Miguel Angel Rosety, Manuel Rosety-Rodriguez.

**Investigation:** Rosalia Fernandez Vazquez, Javier Martinez Blanco, Miguel Angel Rosety.

**Methodology:** Jose Ramon Alvero-Cruz, Rosalia Fernandez Vazquez, Javier Martinez Blanco, Ignacio Rosety, Antonio Jesus Diaz, Miguel Angel Rosety, Francisco Javier Ordonez.

**Project administration:** Jose Ramon Alvero-Cruz, Rosalia Fernandez Vazquez, Javier Martinez Blanco.

**Resources:** Javier Martinez Blanco.

**Software:** Ignacio Rosety.

**Supervision:** Jose Ramon Alvero-Cruz, Miguel Angel Rosety.

**Validation:** Francisco Javier Ordonez.

**Visualization:** Antonio Jesus Diaz.

**Writing – original draft:** Jose Ramon Alvero-Cruz, Rosalia Fernandez Vazquez, Ignacio Rosety, Miguel Angel Rosety, Francisco Javier Ordonez.

**Writing – review & editing:** Jose Ramon Alvero-Cruz, Javier Martinez Blanco, Manuel Rosety-Rodriguez, Francisco Javier Ordonez.

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
