## [Decision Letter · Decision Letter 0]

9 Dec 2020

PONE-D-20-36065

Proposed cut-off points for anthropometric and bioelectrical measures based on overweight and obesity criteria in Spanish institutionalised elderly people.

PLOS ONE

Dear Dr. Ordonez,

Thank you for submitting your manuscript to PLOS ONE. After careful consideration, we feel that it has merit but does not fully meet PLOS ONE’s publication criteria as it currently stands. Therefore, we invite you to submit a revised version of the manuscript that addresses the points raised during the review process.

We look forward to receiving your revised manuscript.

Kind regards,

Simone Perna, Ph.D

Academic Editor

PLOS ONE

Additional Editor Comments:

Dear Author

the following article is well written and worth it for the publication. Only minor comments should be addressed in according to the reviewers suggestions.

Journal Requirements:

3. In your Methods section, please provide additional information about the participant recruitment method and the demographic details of your participants. Please ensure you have provided sufficient details to replicate the analyses such as: a) the recruitment date range (month and year), b) a description of any inclusion/exclusion criteria that were applied to participant recruitment, c) a table of relevant demographic details, d) a statement as to whether your sample can be considered representative of a larger population, e) a description of how participants were recruited, and f) descriptions of where participants were recruited and where the research took place.

Reviewers' comments:

Reviewer's Responses to Questions

**Comments to the Author**

1. Is the manuscript technically sound, and do the data support the conclusions?

Reviewer #1: Yes

Reviewer #2: Yes

Reviewer #3: Partly

2. Has the statistical analysis been performed appropriately and rigorously? 

Reviewer #1: I Don't Know

Reviewer #2: Yes

Reviewer #3: I Don't Know

3. Have the authors made all data underlying the findings in their manuscript fully available?

Reviewer #1: Yes

Reviewer #2: Yes

Reviewer #3: Yes

4. Is the manuscript presented in an intelligible fashion and written in standard English?

Reviewer #1: Yes

Reviewer #2: Yes

Reviewer #3: Yes

5. Review Comments to the Author

Reviewer #1: The paper is written in a good English, sections are quite complete in terms of information and discussion and data appear interesting, even if the argument is not particularly novel.

There are some aspects that should be addressed to improve the general quality of the article:

- ROC and AUC abbrevation should be explained also in abstract section

- Girth should be named "circumference" as latest research is generally using this term instead of girth, especially if WC (waist circumference) abbreviation is used.

- Introduction section is too long and should be reduced as the aspects discussed are quite well known and should not described in detail.

- Methods: Issues regarding the measure of waist circumference and height should be largely discussed, also in discussion section: waist circumference is a heavily operator dependent parameter and hard to measure especially in obese subjects due to skinfolds and fat rolls; height in elderly is hard to measure (hunched back) and is commonly approximated.

- BMI: Another sentence should be added discussing BMI classification also in methods section, these references may be useful:

Kvamme, J.M.; Holmen, J.; Wilsgaard, T.; Florholmen, J.; Midthjell, K.; Jacobsen, B.K. Body mass index and mortality in elderly men and women: The Tromsø and HUNT studies. J. Epidemiol. Community Health 2012, 66, 611–617.

Sergi, G.; Perissinotto, E.; Pisent, C.; Buja, A.; Maggi, S.; Coin, A.; Grigoletto, F.; Enzi, G. An adequate threshold for body mass index to detect underweight condition in elderly persons: The Italian Longitudinal Study on Aging (ILSA). Journals Gerontol. - Ser. A Biol. Sci. Med. Sci. 2005, 60, 866–871.

Janssen, I.; Mark, A.E. Elevated body mass index and mortality risk in the elderly. Obes. Rev. 2007, 8, 41–59.

- Line 99, too much data, should be put in results section

- Bioimpedance: Is the instrument validated for measuring trunk fat and visceral fat ratio in particular? add a reference.

- descriptive characteristics show that the mean age in men is 71.5 +- 7.3 y, please check if subjects are all older than 65 years old.

Reviewer #2: The study presented is well conducted with potential clinical impact. The analysis and conclusions are convincing and publication is recommended after a few minor issues are addressed.

In the abstract you need to mention what ROC and AUC stand for. Overall, all acronyms in the manuscript should be defined including BMI (L. 91), WHO (L. 141), CT and DEXA (L. 223).

The sentence starting from L. 63 to L. 81 in the introduction is too long. Please consider rephrasing.

The sentence in L. 81 is incomplete. Please add the word "persons" or "individuals" at the end.

Delete the word "of" in the sentence (L. 89)

The word "size" in L. 117 is incorrectly used in the definition of BMI and should be replaced with "height."

Delete "of" in L. 122.

Significance levels should be mentioned at the foot of all tables.

In the discussion, L.282, the following sentence should be changed: “several limitations to the study should be also recognized” to several limitations to the study should also be recognized.

Reviewer #3: This study is well-conducted and of great interest.

My additional comments:

- The age of subjects reported in the abstract is 74.3 and 71.5, while along the text and in the table it is 84.3 and 81.5. Which are the correct ones?

- The bioelectrical impedance used for the measures of trunk and visceral fat is validated and it is comparable to Dxa measure for visceral vat (core scan software)? You should include the validation study in the methods section.

- The exclusion of patients with waist girth greater than 130 cm could be referred as a limitation.

- I suggest to cite one of these 2 interested studies:

Perna et al. Comparison between Bioimpedance Analysis and Dual-Energy X-ray Absorptiometry in assessment of body composition in a cohort of elderly patients aged 65-90 years. Adv Gerontol

. 2019;32(6):1023-1033.

Spadaccini et al. DXA-Derived Visceral Adipose Tissue (VAT) in Elderly: Percentiles of Reference for Gender and Association with Metabolic Outcomes. Life (Basel). 2020 Aug 24;10(9):163.

6. PLOS authors have the option to publish the peer review history of their article (what does this mean?). If published, this will include your full peer review and any attached files.

Reviewer #1: No

Reviewer #2: No

Reviewer #3: No

---

## [Author Response · Author response to Decision Letter 0]

15 Feb 2021

Reviewer #1: 

The paper is written in a good English, sections are quite complete in terms of information and discussion and data appear interesting, even if the argument is not particularly novel.

There are some aspects that should be addressed to improve the general quality of the article:

Q. ROC and AUC abbrevation should be explained also in abstract section

A. Thanks for this suggestion. We have detailed both items.

Q. Girth should be named "circumference" as latest research is generally using this term instead of girth, especially if WC (waist circumference) abbreviation is used.

A. Thanks. We do agree. It was changed throughout the text.

Q: Introduction section is too long and should be reduced as the aspects discussed are quite well known and should not described in detail.

A. We do agree. Accordingly, we have shortened it

Q. Methods: Issues regarding the measure of waist circumference and height should be largely discussed, also in discussion section: waist circumference is a heavily operator dependent parameter and hard to measure especially in obese subjects due to skinfolds and fat rolls; height in elderly is hard to measure (hunched back) and is commonly approximated.

A. We do understand your concern about this issue. In fact, all anthropometric measures (including both height and WC) were obtained in accordance with the guidelines of the International Society for Advancement in Kinanthropometry and carried out by the same researcher accredited with level 3 and with a technical error of measurement of less than 1% for all measures. Additionally, a sentence addressing this issue has also been included in the discussion section. 

Q. BMI: Another sentence should be added discussing BMI classification also in methods section, these references may be useful:

Kvamme, J.M.; Holmen, J.; Wilsgaard, T.; Florholmen, J.; Midthjell, K.; Jacobsen, B.K. Body mass index and mortality in elderly men and women: The Tromsø and HUNT studies. J. Epidemiol. Community Health 2012, 66, 611–617.

Sergi, G.; Perissinotto, E.; Pisent, C.; Buja, A.; Maggi, S.; Coin, A.; Grigoletto, F.; Enzi, G. An adequate threshold for body mass index to detect underweight condition in elderly persons: The Italian Longitudinal Study on Aging (ILSA). Journals Gerontol. - Ser. A Biol. Sci. Med. Sci. 2005, 60, 866–871.

Janssen, I.; Mark, A.E. Elevated body mass index and mortality risk in the elderly. Obes. Rev. 2007, 8, 41–59.

A. We apologize but we have not fully understood this suggestion. To the best of our knowledge, the classification based on the National Institutes for Health (NIH)/WHO guidelines is still widely used both on research and in clinical practice.

Q. Line 99, too much data, should be put in results section

A. We do agree. Mainly if we take into account these data are available in Table 1. Accordingly, we have shortened it.

Q. Bioimpedance: Is the instrument validated for measuring trunk fat and visceral fat ratio in particular? add a reference.

A. We do agree it was necessary to include a validation study that could support it (Manios et al. 2013)

Q. descriptive characteristics show that the mean age in men is 71.5 +- 7.3 y, please check if subjects are all older than 65 years old.

A. Thanks for this suggestion. The correct ones are the listed in materials and tables. Accordingly, the abstract has been corrected. We apologize for this mistake.

Reviewer #2: 

The study presented is well conducted with potential clinical impact. The analysis and conclusions are convincing and publication is recommended after a few minor issues are addressed.

Q. In the abstract you need to mention what ROC and AUC stand for. Overall, all acronyms in the manuscript should be defined including BMI (L. 91), WHO (L. 141), CT and DEXA (L. 223).

A. Thanks for this suggestions. We have corrected all items.

Q. The sentence starting from L. 63 to L. 81 in the introduction is too long. Please consider rephrasing.

A. We have changed it and shortened it

Q. The sentence in L. 81 is incomplete. Please add the word "persons" or "individuals" at the end.

A. This sentence was removed in order to shorten the section.

Q. Delete the word "of" in the sentence (L. 89)

A. Thanks. It was deleted.

Q. The word "size" in L. 117 is incorrectly used in the definition of BMI and should be replaced with "height."

A. Thanks. It was replaced.

Q. Delete "of" in L. 122.

A. Thanks. It was deleted.

Q. Significance levels should be mentioned at the foot of all tables.

A. It has been included at the foot of all tables.

Q. In the discussion, L.282, the following sentence should be changed: “several limitations to the study should be also recognized” to several limitations to the study should also be recognized.

A. Thanks. It was changed.

Reviewer #3: 

This study is well-conducted and of great interest. My additional comments:

Q. The age of subjects reported in the abstract is 74.3 and 71.5, while along the text and in the table it is 84.3 and 81.5. Which are the correct ones?

A. Thanks for this suggestion. The correct ones are the listed in materials and tables. Accordingly, the abstract has been corrected. We apologize for this mistake.

Q. The bioelectrical impedance used for the measures of trunk and visceral fat is validated and it is comparable to Dxa measure for visceral vat (core scan software)? You should include the validation study in the methods section.

A. We do agree it was necessary to include a validation study that could support it (Manios et al. 2013)

Q. The exclusion of patients with waist girth greater than 130 cm could be referred as a limitation.

A. Thanks for that suggestion. We have reported it as a new limitation of the current study.

Q. I suggest to cite one of these 2 interested studies:

Perna et al. Comparison between Bioimpedance Analysis and Dual-Energy X-ray Absorptiometry in assessment of body composition in a cohort of elderly patients aged 65-90 years. Adv Gerontol. 2019;32(6):1023-1033.

Spadaccini et al. DXA-Derived Visceral Adipose Tissue (VAT) in Elderly: Percentiles of Reference for Gender and Association with Metabolic Outcomes. Life (Basel). 2020 Aug 24;10(9):163.

A. Thank for your recommendation. We do agree both references could be of great interest for our potential readers.

---

## [Decision Letter · Decision Letter 1]

18 Feb 2021

Proposed cut-off points for anthropometric and bioelectrical measures based on overweight and obesity criteria in Spanish institutionalised elderly people.

PONE-D-20-36065R1

Dear Prof. Ordonez,

We’re pleased to inform you that your manuscript has been judged scientifically suitable for publication and will be formally accepted for publication once it meets all outstanding technical requirements.

Kind regards,

Prof Simone Perna, Ph.D

Academic Editor

PLOS ONE

Additional Editor Comments (optional):

Dear Author

greetings,

After your careful revision, this article is valuable to be accepted on Plos One.

Thanks a lot for your cooperation

Reviewers' comments:

Reviewer's Responses to Questions

**Comments to the Author**

1. If the authors have adequately addressed your comments raised in a previous round of review and you feel that this manuscript is now acceptable for publication, you may indicate that here to bypass the “Comments to the Author” section, enter your conflict of interest statement in the “Confidential to Editor” section, and submit your "Accept" recommendation.

Reviewer #1: All comments have been addressed

Reviewer #2: All comments have been addressed

Reviewer #3: All comments have been addressed

2. Is the manuscript technically sound, and do the data support the conclusions?

Reviewer #1: Yes

Reviewer #2: Yes

Reviewer #3: Yes

3. Has the statistical analysis been performed appropriately and rigorously? 

Reviewer #1: Yes

Reviewer #2: Yes

Reviewer #3: Yes

4. Have the authors made all data underlying the findings in their manuscript fully available?

Reviewer #1: Yes

Reviewer #2: Yes

Reviewer #3: Yes

5. Is the manuscript presented in an intelligible fashion and written in standard English?

Reviewer #1: Yes

Reviewer #2: Yes

Reviewer #3: Yes

6. Review Comments to the Author

Reviewer #1: (No Response)

Reviewer #2: The authors have replied to all my comments raised during the revision of the manuscript. I have no further comments.

Reviewer #3: The authors have met all the requirements.

The article is suitable for publication without any additional corrections.

7. PLOS authors have the option to publish the peer review history of their article (what does this mean?). If published, this will include your full peer review and any attached files.

Reviewer #1: No

Reviewer #2: No

Reviewer #3: No

---

## [Editor Report · Acceptance letter]

26 Feb 2021

PONE-D-20-36065R1 

Proposed cut-off points for anthropometric and bioelectrical measures based on overweight and obesity criteria in Spanish institutionalised elderly people.

Dear Dr. Ordonez:

I'm pleased to inform you that your manuscript has been deemed suitable for publication in PLOS ONE. Congratulations! Your manuscript is now with our production department. 

Kind regards, 

on behalf of

Professor Simone Perna 

Academic Editor

PLOS ONE